# Individual- and Connectivity-Based Real-Time fMRI Neurofeedback to Modulate Emotion-Related Brain Responses in Patients with Depression: A Pilot Study

**DOI:** 10.3390/brainsci12121714

**Published:** 2022-12-14

**Authors:** Maximilian Maywald, Marco Paolini, Boris Stephan Rauchmann, Christian Gerz, Jan Lars Heppe, Annika Wolf, Linda Lerchenberger, Igor Tominschek, Sophia Stöcklein, Paul Reidler, Nadja Tschentscher, Birgit Ertl-Wagner, Oliver Pogarell, Daniel Keeser, Susanne Karch

**Affiliations:** 1Department of Psychiatry and Psychotherapy, University Hospital LMU Munich, Nußbaumstr. 7, 80336 Munich, Germany; 2Department of Radiology, University Hospital LMU Munich, Marchioninistraße 15, 81377 Munich, Germany; 3Psychosomatic Day Care Unit Westend, Westendstraße 185, 80686 Munich, Germany

**Keywords:** real-time fMRI neurofeedback, depression, insula, dorsolateral prefrontal cortex, Responder, Non-Responder, NEO-FFI

## Abstract

Introduction: Individual real-time functional magnetic resonance imaging neurofeedback (rtfMRI NF) might be a promising adjuvant in treating depressive symptoms. Further studies showed functional variations and connectivity-related changes in the dorsolateral prefrontal cortex (dlPFC) and the insular cortex. Objectives: The aim of this pilot study was to investigate whether individualized connectivity-based rtfMRI NF training can improve symptoms in depressed patients as an adjunct to a psychotherapeutic programme. The novel strategy chosen for this was to increase connectivity between individualized regions of interest, namely the insula and the dlPFC. Methods: Sixteen patients diagnosed with major depressive disorder (MDD, ICD-10) and 19 matched healthy controls (HC) participated in a rtfMRI NF training consisting of two sessions with three runs each, within an interval of one week. RtfMRI NF was applied during a sequence of negative emotional pictures to modulate the connectivity between the dlPFC and the insula. The MDD REAL group was divided into a Responder and a Non-Responder group. Patients with an increased connectivity during the second NF session or during both the first and the second NF session were identified as “MDD REAL Responder” (N = 6). Patients that did not show any increase in connectivity and/or a decreased connectivity were identified as “MDD REAL Non-Responder” (N = 7). Results: Before the rtfMRI sessions, patients with MDD showed higher neural activation levels in ventromedial PFC and the insula than HC; by contrast, HC revealed increased hemodynamic activity in visual processing areas (primary visual cortex and visual association cortex) compared to patients with MDD. The comparison of hemodynamic responses during the first compared to during the last NF session demonstrated significantly increased BOLD-activation in the medial orbitofrontal cortex (mOFC) in patients and HC, and additionally in the lateral OFC in patients with MDD. These findings were particularly due to the MDD Responder group, as the MDD Non-Responder group showed no increase in this region during the last NF run. There was a decrease of neural activation in emotional processing brain regions in both groups in the last NF run compared to the first: HC showed differences in the insula, parahippocampal gyrus, basal ganglia, and cingulate gyrus. Patients with MDD demonstrated deceased responses in the parahippocampal gyrus. There was no significant reduction of BDI scores after NF training in patients. Conclusions: Increased neural activation in the insula and vmPFC in MDD suggests an increased emotional reaction in patients with MDD. The activation of the mOFC could be associated with improved control-strategies and association-learning processes. The increased lOFC activation could indicate a stronger sensitivity to failed NF attempts in MDD. A stronger involvement of visual processing areas in HC may indicate better adaptation to negative emotional stimuli after repeated presentation. Overall, the rtfMRI NF had an impact on neurobiological mechanisms, but not on psychometric measures in patients with MDD.

## 1. Introduction

Depression is a mental disorder that comprises diverse symptoms such as sadness, loss of joy, social withdrawal, insomnia, concentration deficits, low self-esteem. Subsequently, psychosocial functioning can be at risk [1]. More than 164 million people suffer from depression worldwide; depression ranked among the top five non-fatal illnesses in terms of years lived with disease (YLD) in 2017 [2].

Various different interventions exist for the treatment of depression, e.g., psychological and psychopharmacological therapies and the combination of those. Several meta-analyses and randomized controlled trials have provided evidence that these interventions are effective in the acute treatment of depression [3,4,5]. However, there is also some evidence that a relevant number of patients suffer from persistent symptoms [6], and therapeutic interventions often cannot prevent recurrence in major depression [7]. For instance, Vittengl et al. [8] showed a relapse/recurrence rate of 29% in the first year, and 54% in the second year. In addition, 30% to 50% of those considered to be remitted still struggle with residual depressive symptoms [9]. Therefore, the need for improved treatment approaches is essential in order to reduce depressive symptoms.

Numerous studies demonstrated altered brain responses in patients with depression. A major working group focusing on cortical structural alterations in patients diagnosed with major depressive disorder (MDD) reported the results of 2149 patients with MDD and 7957 healthy controls. Results demonstrated thinner cortical grey matter in adult patients with MDD compared to controls in the orbitofrontal cortex (OFC), anterior and posterior cingulate, insula and temporal lobes. In addition, adolescents with MDD had a lower total surface area and regional reductions in frontal regions (e.g., medial OFC and superior frontal gyrus), as well as in primary and higher-order visual, somatosensory, and motor areas [10].

A quantitative meta-analysis on functional brain changes in patients with depression, compared to healthy subjects, showed that a specific network of brain regions is associated with the pathophysiology of depression, including decreased responses in the frontal and temporal cortex as well as the insula and cerebellum [11]. An increase of brain activity has been demonstrated in these brain structures under antidepressive medication [11].

In large meta-analytic studies, loss of grey matter in the dACC and right and left insula is evident across all diagnoses [12]. These regions of the so-called salience network appear to be potential markers of behavioural processing and dysregulation in psychiatric disorders [12,13,14]. Anomalies in these areas have also been found in large meta-analyses regarding brain volumes, DTI, resting-state functional MRI connectivity (rsfcMRI), and task-based fMRI [12,15,16,17]. Changes in these networks have also been associated with early childhood maltreatment, thus linking early negative experiences to current depressive symptoms [18].

RtfMRI NF is a new non-invasive technique that is based on the assumption that brain activity can be modulated with the aid of direct feedback on the neural processes within specific brain regions and/or the connectivity between brain regions. It is assumed that neurofeedback (NF) induced brain modulation can lead to altered perceptions, cognitive functions, and behavioural responses [19]. There are different theories about the underlying mechanisms of NF. Various authors assume that NF follows the principles of operant conditioning. Bray et al. [20] interpret fMRI-NF’s mechanisms as instrumental conditioning, because in their paradigm neural activity in a specific brain region could be rewarded directly without overt behavior..

RtfMRI NF studies have shown that the self-regulation of different brain regions, which are associated with emotional processing, is possible, e.g., the amygdala [21,22,23], the insula [22,24,25] and the subgenual part of the ACC [26]. In addition, growing evidence suggests that modulation of brain responses with the aid of rtfMRI NF is possible in clinical populations, e.g., in patients with tobacco [27] and alcohol use disorders [28], obsessive compulsive disorder [29], anxiety [30,31,32], and depression [33,34].

Several different target regions and NF strategies have been used to modulate brain responses in patients with major depressive disorder in order to improve clinical effectiveness. Linden, Habes, Johnston, Linden, Tatineni, Subramanian, Sorger, Healy, and Goebel [33] demonstrated the upregulation of activity in brain-areas associated with positive emotions, e.g., the ventrolateral PFC, the insula, the dlPFC, the medial temporal lobe, and the OFC in medicated patients with depression during the presentation of emotional pictures. The treatment group showed a clinically relevant improvement according to the Hamilton-Depressive-Rating-Scale (HDRS); the control group, which participated in the same experimental paradigm without NF, did not show any symptom improvement. Takamura, Okamoto, Shibasaki, Yoshino, Okada, Ichikawa, and Yamawaki [34] examined the antidepressant effect of NF training for the left dlPFC activity. Six patients participated in the study and were instructed to increase the activity within the dlPFC. The depression scores were significantly reduced after five days of NF training as compared to before.

A double-blinded, placebo-controlled randomized clinical study of Young et al. [35] showed an increase in the hemodynamic response of the amygdala to positive stimuli, i.e., positive autobiographic memories, after two rtfMRI NF sessions in unmedicated depressed patients, compared to a baseline and to the control group. In addition, the experimental group showed in 12 out of 16 patients a decrease of 50% in the Montgomery-Åsberg Depression Rating Scale (MADRS) score meeting criteria of remission, compared with two persons in the control group.

Mehler et al. [36] conducted a single-blinded, randomized study on medicated patients with major depression. They separated the sample of 43 patients into a treatment-group receiving rtfMRI NF in a region (limbic and frontal) associated with emotion regulation, and a control-group receiving five sessions rtfMRI NF in a control region (parahippocampal area) involved in the processing of higher visual information (scene, face, and animal pictures). Across groups there was a 43% improvement on HDRS and 38% of the patients remitted. Due to the lack of a sham-control group no answer can be given on whether the rtfMRI induced improvements were specific to particular brain regions.

The review of Young et al. [37] gathers information about amygdala-associated rtfMRI NF training in patients with major depressive disorder and healthy subjects. Overall, the results of various studies concerning amygdala-based NF training are encouraging and suggest the clinical potential of this method in alleviating symptoms of major depressive disorder.

A very recent study of Ahrweiler et al. [38], with adolescents suffering from depression, focused on the modulation of amygdala-hippocampal activity during adolescents viewing their own happy faces and recalling a positive autobiographical memory. The results indicated that reduced depression was related to enhanced activation in brain regions related to emotional regulation and cross-modal areas during a self-recognition task. The authors concluded that these results may indicate that NF can induce short-term neurobiological changes in the self-referential and emotional regulation networks associated with reduced symptom severity. In summery, rtfMRI NF might be a good adjuvant to established treatment methods for depression. However, further evidence is needed on the specific brain-regions and NF strategies that lead to the best outcome within a minimum of NF sessions. For example, the connectivity feedback between multiple brain regions might be even more powerful than single ROI approaches concerning the clinical effectiveness [39,40]. One study reported a reduced connectivity between dlPFC and insula in the right hemisphere in depressive patients compared to a healthy control group (Kandilarova et al. [41]). The authors assumed that the connection between these two structures could be either part of the ventral fronto-parietal control network (FPN), which is involved in the bottom-up attention control, or of the dorsal FPN, associated with bottom-up attention regulation. Another explanation could be that these brain regions are part of the salience network (SN). Some evidence exists for a hypoconnectivity in depressive patients within the FPN [42] and also within the SN, especially in the right anterior insula [43]. Furthermore, these findings could be part of the underlying pathophysiological mechanisms in depression, which are often accompanied by deficits in attention control and decision-making [41]. In general, the dlPFC is associated with cognitive and executive functions, e.g., attention control, working memory, planning, abstract thinking, and/or goal-oriented acting [44,45,46]. However, there is also evidence that this region is involved in emotion-regulation using cognitive strategies [47,48], e.g., to suppress sadness [49] or anxiety [50]. Studies have shown that the dlPFC’s activity and metabolism were reduced in patients with depression compared to a healthy control group [51,52,53].

In patients with depression, Stratmann et al. [54] detected a reduction in grey matter volume in the hippocampus and the insular cortex compared to a healthy control group. Grey matter volume was negatively correlated with depressive symptoms [55]. The insular cortex is involved in different sensorimotor, olfactory and gustatory, cognitive and social-emotional processes. Thus, it has an integrative function [56], including perceptions of feelings [57], or empathy [58]. Hamilton et al. [59] described in their meta-analysis a higher activation of the insular cortex in depressive patients compared to healthy subjects during processing of affective content. A higher activation was correlated with stronger rumination symptoms [60]. One explanation of these phenomena could be that a stronger insular cortex activation goes together with higher attention and processing of emotional information [61]. There is evidence suggesting that higher insula activation normalizes with pharmacotherapy [62]. While watching negative pictures from the International Affective Picture System (IAPS), depressive patients showed decreased activity in the right insula as well as in the right hippocampus. In contrast, positive pictures induced decreased activity in the left insula as well as in the right ACC in depressed patients, compared to a healthy control group. During the exposure of negative pictures, severity of depression symptoms correlated positively with responses in the left insula, left amygdala, and inferior OFC [63].

Given the functional importance of the insula and the dlPFC in emotion processing, as well as the evidence on their altered functional connectivity in patients with depression, both areas seem promising targets for a connectivity based rtfMRI NF therapy approach. The aim of the present project was to determine whether it is possible to increase the connectivity between these regions in two sessions of rtfMRI NF training within two weeks, in order to influence and normalize bottom-up and top-down processing. We focused especially on the question whether there are any brain activity differences appearing between the beginning and the end of our NF training, compared to a sham NF control group, which watched neurofeedback session video clips of other patients, and a healthy control group, which received the real condition. The influence of rtfMRI NF training on clinical symptomatology has been assessed. To our knowledge, none of the previous studies in patients with major depression applied a connectivity-based NF training in the context of therapy within a psychosomatic day clinic.

First, it was hypothesized that real-time fMRI neurofeedback training leads to specific changes in neural activation of emotion-associated areas, like the insula, amygdala, hippocampus, and basal ganglia (H1).

Second, it was hypothesized that there are differences in brain activity between patients with MDD and healthy subjects. We predicted that patients show enhanced activations in the emotion-associated areas like the insula, amygdala, hippocampus, and basal ganglia, compared to healthy subjects (H2).

Third, it was hypothesized that the neurofeedback training leads to an improvement in clinical symptoms on a subjective and neurobiological level (H3).

## 2. Materials and Methods

### 2.1. Subjects

Our pilot study comprised the investigation of 49 participants, 30 patients with a major depressive disorder (ICD-10 diagnosis). All patients were recruited in the day clinic Westend in Munich, Germany. Key inclusion criteria were age between 18 and 65 years, and the ICD-10 diagnosis of a moderate depressive episode (F 32.1). Exclusion criteria were the presence of structural brain pathologies associated with prior head injury or neoplasm, lifetime diagnosis of psychosis or dependence disorder, pregnancy, lactation, or typical MRI contra-indications like claustrophobia or ferromagnetic implants.

All patients were randomized with a computer algorithm in a real NF training group (MDD REAL; N = 16; ♀ = 11, ♂ = 5; age: M = 33.13, SD = 12.36) and a sham NF group (MDD SHAM; N = 14) (https://www.ultimatesolver.com/de/zufall-gruppen (accessed on 15 November 2022). HC also received a real NF training (HC REAL; (♀ = 10, ♂ = 9; age: M = 24.35, SD = 3.06 (Table 1). Eleven participants had to be excluded from the study because of premature termination (1 × SHAM, 1 × REAL, 2 × HC) or technical problems (4 × REAL, 3 × HC). Unfortunately, due to too many technical problems, the sample of the sham condition turned out to be too small for further analysis. Therefore, the present article focuses on the results of the real condition. The functional MRI results refer to the data of 11 subjects in the REAL MDD group and 14 subjects in the HC REAL group.

The study received approval from the local ethics committee of the Medical Faculty of LMU Munich (237-12) and was designed in accordance with the Declaration of Helsinki and subsequent revisions. Informed consent was obtained from all subjects involved in the study. The participation in the rtfMRI NF training was compensated with €60 per session. Patients participated in the therapeutic programme of the day clinic Westend, Munich. RtfMRI NF training was provided twice as an add-on to the therapy programme and was carried out in an interval of 1–2 weeks. Standardised questionnaires were used in order to assess sociodemographic data, information about depression, personality, and intelligence.

The study was double blinded as subjects did not know which condition they belonged to before and after the NF training. In addition, during each NF session, two investigators were present: one for the fMRI acquisitions, the other one for the communication with the subjects. Investigators that communicated with the subjects were not informed about the group assignment. There was no further query as to whether they knew their condition. During the real condition, responses within the insula or the DLPFC were presented simultaneously with the negative emotional pictures. During the sham condition, participants watched a NF video clip from another person believing that this was their own choice. The healthy control group underwent the real condition.

### 2.2. Psychometric Questionnaires

Different psychometric tests were used to screen for depression, intelligence, and personality traits. The depression symptomatology of the participants was determined using the Beck Depressions Inventory (BDI) [64]. We assessed verbal intelligence with the aid of the verbal intelligence test (WST) [65]. Personality traits were measured using the NEO-Five Factor Inventory (NEO-FFI) [66].

### 2.3. Paradigm

FMRI measurements took place at the Department of Radiology, Ludwig-Maximilian-University of Munich. Before and after each fMRI session, participants’ depressive symptomatology was determined with the BDI. At their first appointment, participants completed the WST and the NEO-FFI. During the NF training, the pulse rate was measured with a pulse oximeter. The visual stimulation consisted of 40 neutral and 40 negative emotional pictures. All pictures originated from the International Affective Picture System (IAPS, http://csea.phhp.ufl.edu (8 March 2021). Negative emotional pictures showed depression-related pictures, e.g., a crying child or a graveyard, the neutral pictures presented, e.g., landscapes or objects like a pencil. The following measurements were acquired during the fMRI session: (1) emotion-associated task, (2) resting state, and (3) rtfMRI NF paradigm (Figure 1). In detail:

(1) Emotion-associated task: The cue exposure paradigm was used as functional localizer. Forty neutral and forty negative emotion-related pictures were presented block-wise using the software program PsychoPy (v1.78.00, [67]). A single run consisted of nine blocks of 40 s each; during five blocks, only neutral pictures were presented, during four blocks, only negative emotion-related pictures were presented. Blocks with neutral pictures and blocks with emotion-related pictures alternated, starting with a block of neutral pictures. Each picture was shown for 1 second. The pictures were presented in the identical sequence in both sessions. Patients were instructed to look at the pictures.

Hemodynamic responses during negative emotional cues and neutral pictures were identified and compared using the multiplanar activation maps calculated in the Turbo-BrainVoyager Version 3.0 (TBV) (Brain Innovation, Maastricht, The Netherlands) online analysis. All computations required to process a functional volume (3D image created from all slices arriving within a TR) are completed by TBV in less than one second. This includes real-time pre-processing such as 3D motion correction, spatial Gaussian smoothing, and temporal filtering [68]. During the functional localizer sequence, the activation cluster with the most extensive BOLD response to negative emotion-related information in the dlPFC and the insula was defined as a region of interest (ROI) for each person and day individually (threshold t = 3). The dlPFC and the insula were identified on the first acquired EPI image of the online analysis using conventional neuroanatomical MRI landmarks [69] plus the multiplanar reconstructions offered by TBV, and later validated in the offline analysis after transfer to Talairach space. With the exception of one person, the individual ROIs were always located on the left hemisphere. The size of the ROIs could vary inter-individually. 

(2) Resting state: Resting state-sequences were acquired each day before and after the NF-task. Participants were instructed to keep their eyes closed without falling asleep and to think of nothing in particular. The results of these sequences will be presented elsewhere (Rauchmann et al., unpublished).

(3) RtfMRI NF-paradigm: The NF-training consisted of two sessions of NF training with three NF runs each. Apart from the NF-tasks during depression-related cues, the paradigm of a single NF run was identical to the paradigm of a cue exposure run. During the presentation of negative emotion associated stimuli, participants were instructed to increase their individual connectivity between the target ROIs. The idea was that patients should individually find their own strategies to influence brain activity. Before the first neurofeedback session, the patients received instruction about brain mechanisms, how neurofeedback works theoretically, and various examples of how brain activity could be improved. All participants were encouraged to apply various strategies to identify the best individual method. Participants were not instructed to use a specific strategy for modulation. However, it was recommended that they should try methods, which in the past used to be successful in coping with negative emotional situations.

Connectivity-based BOLD responses were calculated and visualized using the TBV and the “Network Access Interface” for TBV. This online calculation was based on Pearson’s correlation coefficient and a sliding window of 20 TRs. The connectivity between the target ROIs were visualized using a ‘graphical thermometer’. The thermometer and the pictures were presented side by side during the whole NF training session (see Figure 1).

During the neutral condition, participants were requested to look at the pictures without any further instruction. Between NF runs, participants were asked about their perceived success during the rtfMRI NF training runs. They did not receive any specific feedback regarding their individual performance between the NR runs. However, there was motivational encouragement to continue trying to modulate brain responses and to lie as still as possible.

### 2.4. MRI and fMRI Data Acquisition

Imaging was performed in a 3 Tesla Philips MR System Ingenia scanner with echo planar capability (Release 4.1 Level 3 2013-04-05, Philips Medical Systems Eindhoven Nederland B.V.) and a 32-channel phased array head coil. We acquired a T1-weighted high-resolution 3D data set for each subject for anatomical referencing (Field of View: 240 × 240 × 220; spatial resolution 1 mm isotropic). For functional BOLD imaging during the functional localizer and the NF-paradigm, an EPI sequence was acquired in the same position as the anatomical images (repetition time: 2000 ms; echo time (TE): 30 ms; 25 axial slices; Field of View: 230 × 230 × 132 mm; spatial resolution: 3 × 3 × 4 mm; slice thickness: 4 mm; gap: 0.15 mm).

### 2.5. MRI and fMRI Data Pre- and Post-Processing

We used TBV for initial processing and real-time analysis as well as the feedback for the participants. For further analysis, raw-data in a DICOM-format were converted into a NIfTI-format using MRIConvert (Version 2.0.7 build 369, University of Oregon, Lewis Center for Neuroimaging, 2013). Subsequent post-processing of data and analysis of the fMRI data was carried out by the BrainVoyager software package (Brain Innovation, Maastricht, The Netherlands). The first five images of each run were excluded from any further analysis due to relaxation time effects. The pre-processing of functional data included high-pass filtering (cut-off: three cycles in a time course) to low frequency signal drifts inherent in echo planar imaging, a slice scan time correction, a spatial correction (cubic and trilinear interpolation), spatial smoothing (Gaussian filter with FWHM 4.0 mm), and a 3D motion correction. In addition, functional images were transferred to a standard Talairach brain. Significant BOLD activity was determined by a cross correlation of MR image pixel intensity with an expected hemodynamic response function. Voxelwise *t*-tests were used to identify those brain areas where the signal change was significantly different between emotional-negative responses and neutral stimuli. For each participant, the conditions for negative emotion-relevant pictures and neutral pictures were calculated as regressors. We used the false-discovery-rate correction at a threshold of *p* < 0.001 to counteract the problem of multiple testing. We calculated for the emotion-associated task the contrast negative vs. neutral pictures and compared the groups (MDD REAL; HC REAL) before the NF runs. Furthermore, we compared the first NF run with the last one (negative vs. neutral) once separately for each group and once including both groups as a contrast. Only clusters with a voxel number of more than 30 were reported and visualized. We used a fixed effects analysis because of the relatively small sample size. Fixed effects analyses allow assumptions about effects within the existing sample but do not allow generalizations about the underlying population.

We calculated for the emotion-associated task the contrast negative vs. neutral pictures and compared them to each other before the NF runs. Furthermore, we compared the first NF run with the last one (negative vs. neutral) once separately for each group (MDD REAL; HC REAL) and once including both groups as a contrast.

In addition, the MDD REAL group was divided in two separate groups based on the variation of functional connectivity between the dlPFC and the insula during NF training. A decision was not made according to a specific cut-off or average values, but according to a relative difference in connectivity between NF1 and NF3 on day 1 and day 2, respectively. Patients that could increase their connectivity during the second NF session or during both the first and the second NF session were identified as “MDD REAL Responder” (N = 6). Patients that did not show any increase in connectivity and/or a decreased connectivity were identified as “MDD REAL Non-Responder” (N = 7). We compared the neurobiological responses of the first NF run with the last one (contrast: NF6 vs. NF1, negative vs. neutral) within groups (MDD REAL Responder; MDD REAL Non-Responder).

### 2.6. Statistical Analysis of Psychometric Data

Statistical analysis of the questionnaire ratings of patients and the HC group were calculated via SPSS version 26 with a level of significance *p* < 0.05. Normal distribution was calculated with the Kolmogorov–Smirnov test. Because of the small sample size and missing normal distribution in some tests of the HC group (BDI, NEO-FFI V, WST), we first calculated the non-parametric Mann–Whitney-U test for independent samples or the Wilcoxon test for dependent samples. In a second step, the two-tailed *t*-test for independent or dependent samples was calculated. Results of parametric tests did not differ in any case from non-parametric results, so the *t*-tests were mentioned instead of the non-parametric tests, because of higher power and better comparability. Because of the confounding of sex and age with depression, an integration of a covariate in the statistical model was not possible. For the correlation between questionnaires and ROIs (dlPFC, insula, thalamus, hippocampus, and amygdala), we used the Kendall’s Tau-b correlation coefficient.

## 3. Results

### 3.1. Comparison of Psychometric Data between HC REAL and MDD REAL

The comparison of MDD REAL compared to the HC REAL did not show any significant differences regarding verbal intelligence (Table 2). On the day of the first rtfMRI NF session, MDD REAL demonstrated a significant higher score in the BDI (*p* ≤ 0.001), on the NEO-FFI subscale Neuroticism (*p* = 0.001), and a significant lower value on the subscale Extraversion (*p* = 0.002) than HC REAL (see Table 2). Pre-post-measurements of BDI scores revealed a significant difference in HC REAL, but values stayed on a non-clinical level (before NF training: M = 1.84; SD = 1.68; after NF training: M = 0.47; SD = 1.43; *p* = 0.001). In MDD REAL, the BDI scores did not differ significantly between before and after rtfMRI NF training (before NF training: M = 23.57; SD = 12.79; after NF training: M = 21.43; SD = 10.31; *p* = 0.235).

After the last rtfMRI NF session, MDD REAL (M = 21.43; SD = 10.31) still showed a significant higher score in the BDI (*p* ≤ 0.001) than the HC REAL (M = 0.47, SD = 1.43).

### 3.2. Correlations 

The MDD REAL group showed significant correlations between questionnaire scores and BOLD activation measured in certain ROIs during the functional localizer (see Table 3). There was no significant correlation between sociodemographic variables (sex, age) and depression scores (see Table 4).

### 3.3. Hemodynamic Responses during the Emotion-Associated Task during the Functional Localizer on Day One: MDD REAL vs. HC REAL

During the emotion-associated task of the functional localizer on day one, HC REAL demonstrated increased responses (negative emotional pictures minus neutral pictures) compared to MDD REAL, especially in brain regions that are associated with the processing of visual information and attention, respectively (e.g., primary visual cortex and visual association cortex). By contrast, MDD REAL revealed higher neural responses in emotional processing areas (e.g., insula, (ventro-)medial prefrontal cortex/ACC [vmPFC, BA9/BA10], superior temporal gyrus; see Figure 2, Table 5).

### 3.4. Comparison of Hemodynamic Responses between the First NF Run on Day One and the Last NF Run on Day Two: MDD REAL and HC REAL

Patients and healthy controls used various strategies in order to achieve a satisfying neurofeedback effect. Some patients reported that they tried strategies for emotion regulation to modulate brain responses. Others used strategies that have been helpful in real life situations in the past. We did not find any systematic differences regarding the regulation strategies used between groups.

At the last NF run of day two, patients of the MDD REAL group and HC demonstrated both an increased hemodynamic activity in the PFC, in right and left vmPFC/OFC (BA10), and in the orbital part of inferior frontal gyrus (OFC, BA47), and HC REAL demonstrated an increased activity in left and right vmPFC/OFC (BA10) compared to the first NF run (see Figure 3, Table 6 and Table 7). 

Furthermore, HC REAL showed a reduced hemodynamic activity in emotion-related (insula/BA13, parahippocampal gyrus [PHG]/BA19, basal ganglia, anterior and posterior cingulate gyrus) and visual processing regions (precuneus, superior occipital gyrus, lingual gyrus/BA18), as well as in prefrontal (dlPFC/BA9) and motoric areas (precentral gyrus) at the last NF run compared to the first one (see Figure 3, Table 6).

Patients of the MDD REAL group reduced their hemodynamic activity in emotional (gyrus parahippocampalis) and visual processing regions (Precuneus, frontal eye field/BA8) as well as in prefrontal (dlPFC/BA9, IFG/BA46) and motoric areas (precentral gyrus/BA6) from the first to the last NF run (see Figure 3, Table 7).

### 3.5. Comparison of Hemodynamic Responses between the First and the Last NF Run: MDD REAL Responder vs. MDD REAL Non-Responder

For this contrast, we chose the Bonferroni correction method of multiple comparison, which is more conservative, because of the small sample size.

At the last NF run of day two, MDD REAL Responder demonstrated an increased activity in the superior/middle/medial/inferior frontal gyrus (BA6/10/47), and right and left ACC (BA32/24) compared to the first NF training (see Figure 4, Table 8).

MDD REAL Non-Responder showed a reduced activity in frontoparietal regions at the last NF run (see Figure 4, Table 9).

### 3.6. Comparison of Hemodynamic Responses between the MDD REAL and HC REAL: First vs. Last NF Run

The HC REAL group showed higher BOLD responses in the middle temporal gyrus compared to the patients of the MDD REAL group from the first to the last NF run. In contrast, MDD REAL showed higher responses in the left and right supramarginal gyrus/BA40, the left parahippocampal gyrus/BA19, and the left lateral globus pallidus/amygdala compared to HC REAL (see Figure 5, Table 10).

## 4. Discussion

The current pilot study explored the effects of connectivity-based rtfMRI NF in patients with depression, relative to a healthy control group. We analysed the neurobiological and neuropsychiatric impact of the rtfMRI NF training with the aim to assess its putative applicability as add-on-therapy, by focusing on functional differences between the first and the last NF training session. In contrast to previous activity-based rt-fMRI NF studies on depression [21,32,33,36,70], we used a connectivity-based NF approach in order to directly increase the connectivity between two specific brain regions, i.e., the DLPFC and the insula. The ROIs were determined for each subject using a functional localizer task with negative emotional pictures. The ROI selection was based on several prior studies demonstrating the key roles of the dlPFC and the insula for emotion processing in depression. Taking into account that functional responses in each brain region can alter during the therapeutic process, target ROIs were defined separately for each training session.

### 4.1. Clinical Outcome of Psychometric Data

The MDD REAL group did significantly differ from the HC REAL group in aspects of depressive symptomatology, neuroticism, and extraversion scores similar to other studies [71,72]. Unfortunately, the combination of a therapeutic program including rtfMRI NF did not significantly reduce depressive symptomatology in the MDD REAL group. This result contradicts our H3 that NF training leads to clinical improvement.

Previous studies found both a significant reduction [33,36] and no change [21] in depression scores. It remains unclear if this is due to training techniques, frequency, stimulus presentation (pictures, positive memory recall), selected brain region(s) (insula, amygdala, dlPFC, vlPFC), or the patient samples. We suppose that the relatively short interval between the two training sessions (one week) could have influenced the small therapeutic effect. However, in our study, there was also a number of patients reporting the feeling of higher self-efficacy or control over negative stimulus reactions directly after NF training, while at the same time they doubt the direct influence on the self-rating questionnaires. However, this aspect was not quantitatively verified.

BDI score correlated significantly with dlPFC responses at the first session. However, this association was no longer significant during the second time point. This may indicate a dissociation between brain responses and the perception of acute depressive symptoms. Neuroticism, e.g., defined by symptoms like uncertainty, melancholy, irritability, and nervousness correlated significantly with insula activity. This may be in line with the observation that these symptoms are often also related to depression.

### 4.2. Functional Imaging Data

#### 4.2.1. Comparison of Hemodynamic Responses of MDD REAL and HC REAL Group during the Emotion-Associated Task before the NF Training

During cue exposure (negative > neutral pictures), patients of the MDD REAL group showed an increased BOLD-signal in the insula and the vmPFC (BA9/BA10) compared to the HC REAL group before rtfMRI NF training. This result supports the assumption of H2, that there are differences in brain activity between MDD and HC group. Insula and the vmPFC are assumed to be involved in emotion processing and play an important role in anxiety disorders and depression [41,51,59,60,61,73]. Both areas can possibly be seen as a neurobiological correlate of an increased emotional reaction on negative emotional stimuli in depressive patients. However, there are conflicting findings regarding increased or decreased brain activation in depressed patients. Differences may result from sample size, medication, severity of depression or experimental paradigms. The vmPFC has connections to the hypothalamus and periaqueductal grey, which represents the autonomic visceral activity of emotions. It has also connections to the ventral striatum and nucleus accumbens, which is associated with motivation and reward. Furthermore, it connects to areas associated with fear-conditioning like the amygdala [51,74,75]. Several studies support the hypotheses that vmPFC plays a role in down-regulating amygdala activity, associated with the reduction of anxiety symptoms e.g. [51,76,77,78]. Studies found an abnormal high brain activation in depression before [73,79] and a reduced response after treatment [51,80,81,82]. Even in healthy persons vmPFC-activation correlates positively with negative emotions [83,84]. In contrast, the HC REAL group revealed increased hemodynamic responses in brain regions that are associated with visual processing (BA17) and attention (BA19), compared to patients [85,86]. Thus, this group may use in relation to the MDD REAL group more visual and attentional neuronal pathways for the processing of negative emotional stimuli before NF training. Moreover, a remarkable similarity across groups was revealed in the online-analyses of the emotion-associated task before the NF training. As stated in the ‘Materials and Methods’ section, almost all ROIs were located in the left hemisphere, i.e. in the left insula and the left dlPFC and not in the respective areas of the right hemisphere - determined us-ing pictures with a negative emotional aspect. The issue of lateralization in the brain’s emotional processing including processing of negative emotional visual stimuli has been conversely discussed in the past decades. Up to now, it is still unclear to which degree and at which level each hemisphere contributes to the processing of emotional information – however, activity in the left hemisphere is thought to be mainly involved in the regulation of negative emotions [87]. Given the consistency of the left-accentuated BOLD response, our paradigm and setting seem to be suitable to reveal possible functional effects via similar functional pathways across subjects and to generate suitable ROIs for NF tasks. More-over, the insula and the dlPFC of the left hemisphere are explicitly known to be affected in patients with MDD [88,89] and to be potential targets for rt-fMRI-NF [90].

#### 4.2.2. Comparison of Hemodynamic Responses between the First and the Last NF Run

Healthy controls as well as patients showed increases in brain responses in the bilateral vmPFC (BA10) in the pre-post-NF-measurement. This is surprising given the results of previous studies on the role of vmPFC in patients with depression [51,80,81,82]. Based on previous evidence, we expected a downregulation of the brain activity in this area. The increased vmPFC-activation in our study over NF runs might speak for an increase in motivation with NF success or enhanced self-referential processing of negative stimuli [51,91]. Another explanation could be that NF may be associated with a more conscious downregulation of provoked emotional responses using better cognitive control strategies for processing of negative emotions in the amygdala [74,76,92].

Indeed, the activation in the pre-post-measurement is located more ventrally than in the cue-exposure task. This region can also be seen as part of the medial part of the OFC. The OFC has reciprocal connections to emotional processing areas like the insula, amygdala, ACC and hippocampus [93,94,95]. On the one hand, OFC is thought to play an important role in the pathophysiology of mental illness, and on the other hand, higher levels of neuronal activation in patients are associated with symptom expression [96,97,98]. In addition, the OFC is associated with cognitive control functions and treatment could lead to an increase of the BOLD signal in this area, which is related to symptom reduction [99]. On this account, Rolls [100] proposes a differentiation between the lateral and medial part of the OFC. In his theory the medial OFC (mOFC) reward-system is reduced in depression, as different studies showed before [101,102]. We argue that subtle changes can occur before phenomenological shifts arise [103,104]. The signal change in the OFC of our study may be seen as a direct effect of the rtfMRI NF training and the OFC may play a crucial role in the rtfMRI NF associated learning process. In that context, an increase of activity in the mOFC could be a hint for NF-associated learning processes in depression patients. Due to the memory function of the OFC, its activation might count as hemodynamic correlate of NF association learning as well in the sense of a bottom up effect, which supports the assumption of H1, that rtfMRI NF leads to specific changes in neural activation of emotion-associated areas and of H3 that neurofeedback training leads to an improvement in clinical symptoms.In the MDD REAL group there was an additional increase in the left IFG/OFC (BA47). This area can most likely be attributed to the non-reward/punishment attractor system described by Rolls [100], which is associated with the lateral part of the OFC. A reduced lOFC activity is comparable to a loss of reinforcers [105] or receiving punishments [101], which could lead to depression symptoms [100]. According to Roll’s theory the lOFC is activated more quickly when one is suffering from depression. The stronger the activation in the lOFC, the stronger the influence on other brain structures such as the dlPFC, which in turn supports a negative bias through top-down attention processes. In the case of the present NF training, the stronger activation of the lOFC could indicate an increased sensitivity of the depression patients to failed NF attempts. A reduced activity in the dlPFC was shown in both groups. This result is contrary to most studies, which report a reduced metabolism before and increased metabolism after therapy [51,80,106,107,108]. This result could indicate that attention to negative stimuli or negative received feedback was reduced in both groups in the sense of a learning effect [100]. Another explanation could be that less cognitive strategies have to be used when dealing with negative stimuli [109]. In addition, habituation effects could have led to a reduced valence of negative stimuli. The healthy controls seemed to be able to reduce its emotional reaction to negative stimuli to a significant greater extent than it was the case in the patient group. This can be attributed in particular to reduced activity in emotional processing brain structures such as the basal ganglia, the insula and cingulate gyrus [59,110,111]. In contrast to the healthy control group, a decrease of the BOLD-signal was only be shown in the parahippocampal gyrus (PHG) in the patients group. The parahippocampal gyrus is known to be involved in memory-function, rumination and depression [60,112,113]. A decrease in activity in the PHG could be an expression of a decreased emotional response to the presented visual stimuli in the course of successful NF training. The present results are consistent with those of other studies [11,114].

#### 4.2.3. Comparison of Hemodynamic Responses between the MDD REAL Responder and MDD REAL Non-Responder: First vs. Last NF Run

It appears that the increased response in the vmPFC and mOFC was particularly due to the MDD REAL Responder group, as the MDD REAL Non-Responder group showed no increase in this region during the last NF run. This supports our assumption that successful neurofeedback was particularly indicated by increased activity in the medial OFC.

#### 4.2.4. Comparison of Hemodynamic Responses between the MDD REAL and HC REAL: First vs. Last NF Run

The MDD REAL group showed stronger responses compared to the control group especially in emotional processing brain structures like parahippocampal gyrus (BA19) and the lateral globus pallidus/amygdala. This result supports the assumption of H2, that there are differences between groups and of H1, that rtfMRI NF leads to specific changes in neural activation of emotion-associated areas. This difference was not observed in the emotion-associated task. Thus, the control group and the patients group demonstrated comparable brain responses in this area before NF training and pictures worked as provocation method [115]. This may indicate that the healthy control group adapted better to negative stimuli than the depressive patients did, which has been suggested by previous studies [116,117]. Moreover, patients demonstrated a stronger response in the supramarginal gyrus (SMG). We hypothesize that the SMG is a neurophysiological correlate of the physiological component of emotions. Thus, higher brain activation could also represent a better mental representation of the physical component of emotion, which might have improved under NF training. Since depressed patients adapt less well to negative emotional stimuli, this could explain why this region responds more strongly in patients than in healthy subjects. In contrast, the healthy control group demonstrated especially a higher BOLD-signal in the right posterior middle temporal gyrus (pMTG) compared to the patients over the NF trainings. This structure plays a certain role in emotional processing, as well as in forming new associations and new concepts through creative thinking [118,119]. It could be argued that the healthy control group was more successful in applying new or more creative NF strategies, as indicated by stronger activation in the MTG. The group difference in the insula in the emotion-associated task was not measurable after NF training anymore. This could mean that an increased connectivity between the dlPFC and the insula through NF training could have influenced the activation level while watching negative pictures in the insula. This may be a hint that strengthening the connectivity between prefrontal and emotional processing brain structures could decrease depression relevant brain activity. However, this difference alone did not affect the patients’ subjective feelings, as no significant differences were detectable in the BDI questionnaire. Further studies are necessary to investigate effects of a connectivity-based NF systematically.

## 5. Limitations

We would like to mention various limitations regarding the interpretation of the results. One of the main limitations is the relatively small sample size: the study has been conceptualized as a pilot study in order to gain more insight into the NF capability in patients with depression and the feasibility of connectivity-based NF in order to improve emotional responses. Unfortunately, we had to exclude several patients due to technical problems. For that reason, the evaluable sample was unexpectedly small. Due to the small sample, a fixed effects analysis was used to calculate brain responses. Fixed effects analyses allow assumptions about effects within the existing sample but do not allow generalizations about the underlying population. For the validation of the results of this present pilot study, further studies with a larger sample size would be helpful.

Due to technical problems, the data of the sham group could not be considered in the manuscript.

Therefore, even if our results are suggesting NF specific effects, we cannot rule out that some of our findings are independent of the targeted NF approach. Studies with bigger sample sizes as well as various control groups that may integrate NF-based and not NF-based strategies would be helpful to further assess the specificity of the results. The inclusion of a control group without feedback during the training would be helpful in order to estimate specific NF effects. During the emotion-associated task and the rtfMRI training, negative emotional pictures as well as neutral pictures were presented repetitively to the participants. The repetitive presentation of pictures might be related to habituation and a reduced brain response at later repetitions. However, the habituation was expected in both groups. The optimal number of NF training sessions is not clear yet. Further studies are needed that focus on a methodical examination of this issue. In the present study, all patients participated in two rtfMRI NF sessions with three runs per session in order to improve the power compared to single session training. Moreover, it must be mentioned that our groups differed in age (*p* = 0.014). It is possible that this variable could have an influence on the BOLD signal. Other limitations include that the subjects were not systematically asked if they knew what condition they were in (sham, real) or which strategies they were using.

## 6. Conclusions

In our specific experimental setting, brain activation patterns of connectivity-based rtfMRT NF between the dlPFC and the insula may indicate a more complex and less typical effect in patients with depression compared to a healthy control group. Patients with depression under behavioural group therapy seemed to be less successful in conducting rtfMRI NF than HC trained with identical rtfMRI NF—the latter showing BOLD changes in emotion processing brain areas, i.e., the ACC, the insula, and the basal ganglia, and in temporal areas associated with creative thinking between the first to the last NF training run. Consistently, on a subjective level, there was no substantial therapy effect measurable in patients with depression. Nevertheless, putative improvements were found on a neurobiological level like a reduced PHG response that may stand for less rumination during the NF training. Contrary to previous studies, patients with a behavioural group therapy combined with rtfMRI NF training, as well as the healthy control group, demonstrated an increased response in the vmPFC/mOFC/ACC and a decreased response in the dlPFC from the first to the last NF training run. We could show that this hemodynamic increase in the MDD REAL group was due to the Responders only. This was probably a direct result of the NF training in terms of motivation effects, better control strategies, or as a hemodynamic correlate of association learning. Future studies are needed using a higher number of subjects and training sessions, as well as a longer treatment interval, especially in order to assess therapeutic effects and the persistence of neuronal effects.

## Figures and Tables

**Figure 1 brainsci-12-01714-f001:**
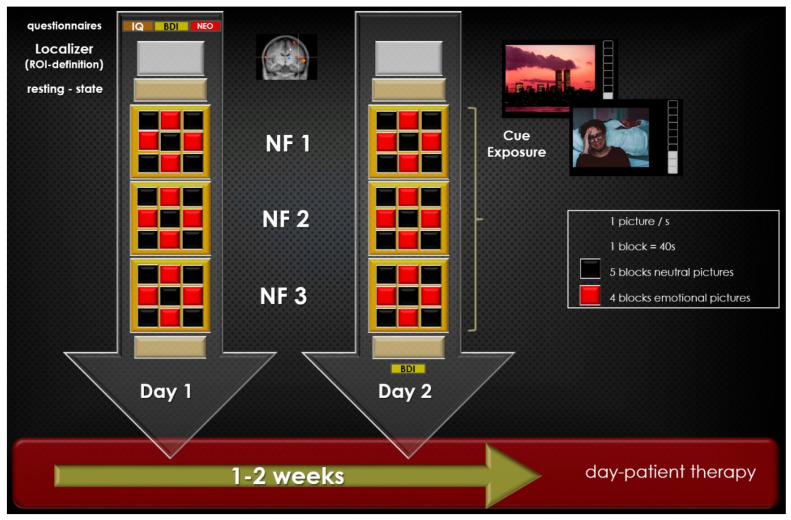
Experimental procedures: the subjects participated in two rtfMRI NF sessions within 2 weeks; in a functional localizer before each NF training, the ROIs insula and dlPFC were determined; during the functional localizer and NF training, neutral and negative emotion-related pictures were shown in blocks of 40 s with 40 pictures of the respective category. Participants were instructed to reduce brain activity during the presentation of the emotion-associated task. During the presentation of neutral information, participants were instructed to simply gaze at the pictures. Before and after each NF-training session, resting-state activity was acquired. NF trainings were embedded in a day-patient therapy. Abbreviations: NF: neurofeedback.

**Figure 2 brainsci-12-01714-f002:**
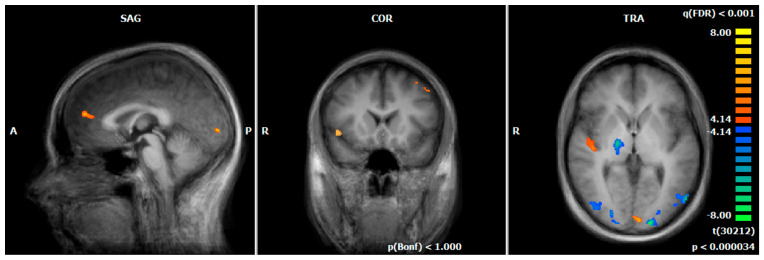
Hemodynamic responses of the emotion-associated task before the NF training (negative emotional pictures > neutral pictures; q (FDR) < 0.001, T-score: −8 to 8, fixed-effects-analysis); in blue, increased responses of the HC REAL group, compared to MDD REAL, e.g., thalamus, primary visual cortex, and visual association cortex; in orange, increased activations of MDD REAL compared to HC REAL, e.g., insula, medial prefrontal cortex, superior temporal gyrus (x = 0; y = 15; z = 3). A = anterior, P = posterior, R = right.

**Figure 3 brainsci-12-01714-f003:**
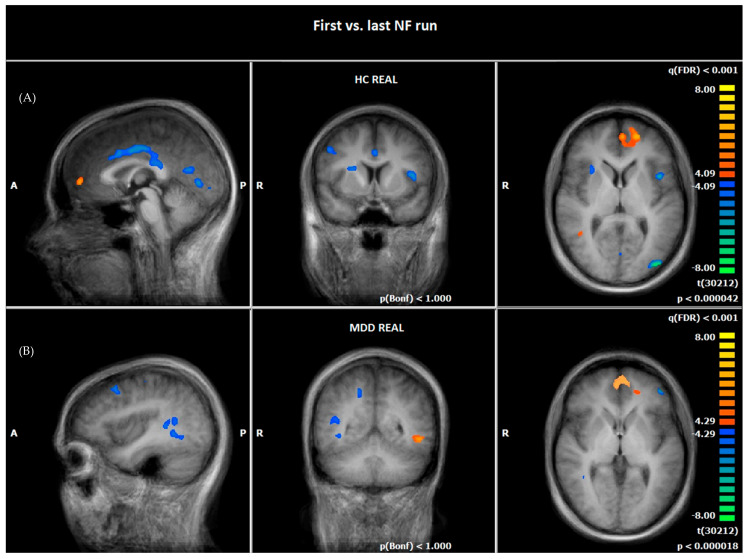
Hemodynamic responses of the first vs. the last NF run (negative emotional pictures > neutral pictures; q (FDR) < 0.001, T-score: −8 to 8, fixed-effects-analysis). (**A**) The HC REAL group demonstrated increased hemodynamic responses, e.g., in the ventromedial prefrontal cortex (orange) and a reduced BOLD-signal (blue) in emotion processing areas, e.g., insula, anterior and posterior cingulate cortex, parahippocampal gyrus, and basal ganglia, after NF training (x = 0; y = 0; z = 7). (**B**) The MDD REAL group demonstrated increased brain responses (orange) in the ventromedial prefrontal cortex and inferior frontal cortex and a reduced BOLD-signal (blue) in emotion processing areas, e.g., parahippocampal gyrus and dlPFC, after NF training (x = 38; y = -50; z = 3). A = anterior, P = posterior, R = right.

**Figure 4 brainsci-12-01714-f004:**
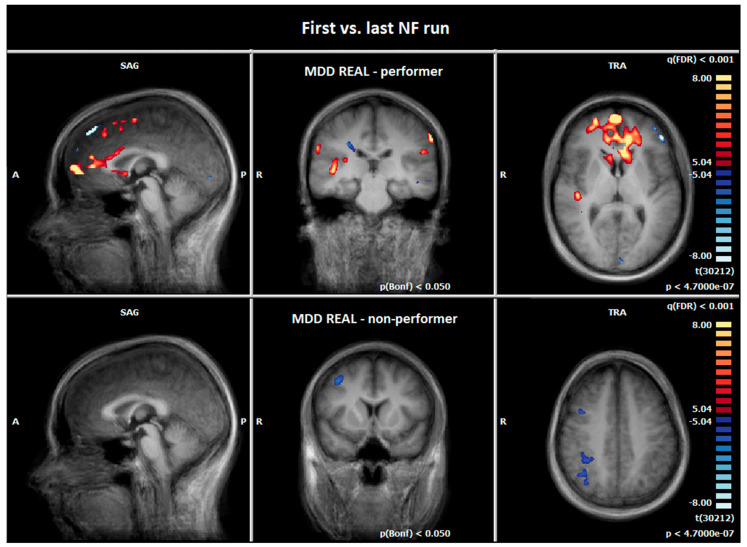
Hemodynamic responses of the first vs. last NF run (negative emotional pictures > neutral pictures; p (Bonf) < 0.05, T-score: −8 to 8, fixed-effects-analysis). The MDD REAL Responder group revealed increased hemodynamic responses at the last NF run, e.g., in the prefrontal cortex and ACC (orange) (x = 0; y = −23; z = 7), the MDD REAL Non-Responder group demonstrated a reduced BOLD-signal (blue) in fronto-parietal regions after NF training (x = 0; y = 10; z = 40). A = anterior, P = posterior, R = right.

**Figure 5 brainsci-12-01714-f005:**
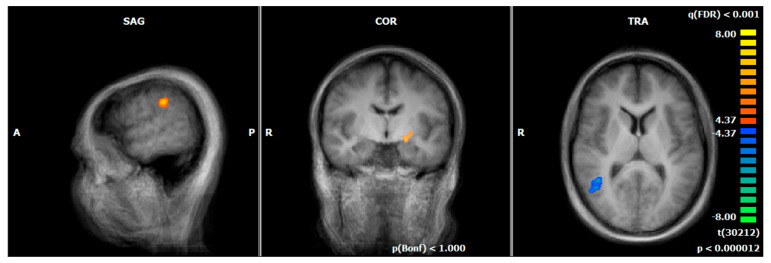
Hemodynamic responses of the first vs. the last NF run: MDD REAL vs. HC REAL (negative emotional pictures > neutral pictures; q (FDR) < 0.001, T-score: −8 to 8, fixed-effects-analysis); in blue, increased hemodynamic responses of the HC REAL group compared to MDD REAL, e.g., in middle temporal gyrus; in orange, increased hemodynamic activations of MDD REAL compared to HC REAL, e.g., supramarginal gyrus and lateral globus pallidus/amygdala (x = 57; y = −8; z = 10). A = anterior, P = posterior, R = right.

**Table 1 brainsci-12-01714-t001:** Age, sex, and medication.

	MDD REAL (N = 16)	HC REAL (N = 19)
Age at entry, M (SD)	33.13 (12.36)	24.35 (3.06)
male, n (%)	5 (31.3)	9 (47.4)
female, n (%)	11 (68.8)	10 (52.6)
*Medication 1, n (%)*		
SSRI	5 (31.3)	
SSNRI	3 (18.8)	
TCA	1 (6.3)	
*Medication 2, n (%)*		
TeCA	1 (6.3)	
atypical AP	1 (6.3)	

Abbreviations: M: mean, SD: standard deviation, SSRI: selective serotonin reuptake inhibitor, SSNRI: serotonin–norepinephrine reuptake inhibitor, TeCA: tetracyclic antidepressant (Mirtazapine), TCA: tricyclic antidepressant (Amitriptyline), AP: antipsychotic (Quetiapin).

**Table 2 brainsci-12-01714-t002:** Comparison of psychometric data between healthy controls (HC REAL) and depressive patients (MDD REAL) before NF.

Questionnaire	HC REAL		MDD REAL		*p*-Value
	M	SD	M	SD	
IQ-Test (WST)	113.47	6.92	107.13	11.87	0.080
NEO-FFI-N	19.00	6.57	33.80	6.44	≤0.001
NEO-FFI-E	29.74	6.70	21.13	8.50	0.002 *
NEO-FFI-N	31.16	7.05	30.93	5.30	0.919
NEO-FFI-A	34.04	6.14	32.07	6.39	0.364
NEO-FFI-C	31.21	6.61	26.73	8.80	0.100
BDI	1.84	1.68	23.33	1.68	≤0.001 *

Abbreviations: M: mean, SD: standard deviation, *: significant.

**Table 3 brainsci-12-01714-t003:** MDD REAL correlations between ROIs and questionnaires.

ROI/ Questionnaire	dlPFC		Insula		Thalamus		Hippocampus		Amygdala	
	r	*p*	r	*p*	r	*p*	r	*p*	r	*p*
NEO-FFI-N	0.32	0.24	0.55	0.04 *	0.10	0.71	0.00	1.00	0.22	0.39
NEO-FFI-E	−0.24	0.39	−0.15	0.60	−0.37	0.21	−0.29	0.29	−0.40	0.14
NEO-FFI-O	−0.15	0.60	0.29	0.29	0.11	0.71	−0.14	0.60	−0.14	0.60
NEO-FFI-A	−0.03	0.91	−0.41	0.13	0.04	0.90	0.26	0.34	0.03	0.92
NEO-FFI-C	−0.55	0.04 *	−0.40	0.14	−0.40	0.17	−0.31	0.25	−0.31	0.25
BDI-pre	0.57	0.04 *	0.37	0.17	−0.14	0.62	−0.11	0.68	0.11	0.68
BDI-post	0.05	0.89	−0.03	0.93	0.11	0.74	0.14	0.68	−0.30	0.36

Abbreviations: r: Kendall’s Tau-b correlation coefficient; *p*: probability-value; *: significant.

**Table 4 brainsci-12-01714-t004:** MDD REAL-Responder correlations between sex/age and questionnaires.

Questionnaire	Sex		Age	
	r	*p*	r	*p*
BDI-pre	0.14	0.70	−0.20	0.54
BDI-post	0.37	0.36	−0.20	0.57

Abbreviations: r: Kendall’s Tau-b correlation coefficient; *p*: probability-value

**Table 5 brainsci-12-01714-t005:** Hemodynamic responses during the emotion-associated task of the functional localizer on day one (negative emotion-associated pictures minus neutral pictures; clusters of >30 voxels, q (FDR)< 0.001, T-score: 8 to −8).

			Centre of Gravity			Size	T-Score	
Brain Region	Side	BA	x	y	z		Ø	Max
HC REAL > MDD REAL								
Middle Frontal Gyrus	L	6	−28	−11	47	577	4.80	5.92
Inferior Frontal Gyrus	R	46	37	33	13	611	4.82	6.97
Precentral Gyrus	L	6	−44	2	33	1118	4.77	5.99
Thalamus	R	−	16	−14	3	655	5.13	7.12
Cuneus	R	17	21	−88	6	419	5.11	6.86
Middle Occipital Gyrus	R	19	36	−74	9	992	5.03	7.72
Middle Occipital Gyrus	L	19	−36	−76	5	3843	5.21	9.21
MDD REAL > HC REAL								
Middle Frontal Gyrus	R	8	30	21	46	665	4.80	6.56
	L	8	−33	18	48	476	4.82	5.95
Medial Frontal Gyrus	L	9	0	45	17	260	4.99	6.86
Thalamus	L	−	−9	−31	15	482	4.75	6.22
Postcentral Gyrus	R	2	50	−18	27	584	4.82	6.08
Precentral Gyrus	R	6	54	−9	38	288	4.67	5.88
Posterior Insula	R	13	46	−11	1	517	4.58	5.90
Anterior Insula	R	13	46	11	−5	339	4.92	6.30
Superior Temporal Gyrus	R	22	36	−57	21	251	4.76	6.09
	L	41	−52	−27	16	244	4.84	6.38
Middle Temporal Gyrus	L	39	−40	−51	9	644	4.56	5.63
Supramarginal Gyrus	R	40	56	−345	22	275	5.10	6.97
Lingual Gyrus	L	17	−4	−90	2	267	5.46	8.02
Cuneus	L	18	−8	−71	18	265	4.59	5.40

Abbreviations: BA: Brodmann area; side: hemisphere; x: Talairach coordinate x-axis; y: Talairach coordinate y-axis; z: Talairach coordinate z-axis; max: maximal T-score; Ø: average t-score; size: cluster size; voxels: number of activated voxels; L: left; R: right.

**Table 6 brainsci-12-01714-t006:** Comparison of hemodynamic responses between the first and the last NF run, HC REAL and MDD REAL (negative emotion-associated pictures minus neutral pictures; clusters of >30 voxels, q (FDR) < 0.001, T-score: 8 to −8).

HC REAL								
			Centre of Gravity			Size	T-Score	
Brain Region	Side	BA	x	y	z		Ø	Max
(A) Last NF run vs. first NF run								
Medial Frontal Gyrus	L	10	−8	48	7	1562	4.84	7.35
Middle Temporal Gyrus	R	39	42	−56	10	305	4.60	5.62
(B) First NF run vs. last NF run								
Middle Frontal Gyrus	R	9	33	36	28	2296	−4.83	−6.71
	R	9	46	7	37	243	−4.51	−5.33
Precentral Gyrus	L	6	−43	−3	50	316	−5.15	−6.81
Cuneus	L	18	−17	−67	18	1140	−4.58	−5.78
Precuneus	R	31	16	−64	20	1427	−4.85	−6.75
	L	19	−28	−72	30	286	−4.67	−6.24
Lingual Gyrus	L	17	−19	−87	−2	607	−4.78	−6.07
	L	18	−2	−76	0	492	−4.55	−5.90
Middle Occipital Gyrus	L	19	−36	−80	10	794	−5.41	−7.85
Supramarginal Gyrus	L	40	−56	−43	33	669	−4.71	−5,99
	R	40	52	−42	30	2122	−4.72	−6.62
Superior Occipital Gyrus	R	19	34	−72	21	1489	−4.59	−5.851
Cingulate Gyrus	L/R	23/24	−1	−14	31	3184	−4.74	−6.13
Posterior Cingulate Gyrus	L/R	23/31	−1	−63	16	261	−4.66	−5.72
Insula	R	13	29	16	17	1232	−4.68	−5.97
	L	13	−40	9	9	521	−5.06	−6.76
Claustrum	R	−	30	15	5	246	−4.47	−5.01
Parahippocampal Gyrus	R	19	23	−50	−2	603	−4.48	−5.66
	R	19	−31	−46	−5	4549	−4.86	−7.57
Globus Pallidus	L	−	−16	−8	−2	1305	−4.66	−6.34
Medial Globus Pallidus	L	−	−15	−10	−2	1467	−4.62	6.34

Abbreviations: BA: Brodmann area; side: hemisphere; L: left; R: right; x: Talairach coordinate x-axis; y: Talairach coordinate y-axis; z: Talairach coordinate z-axis; max: maximal T-score; Ø: aver-age T-score; size: cluster size; voxels: number of activated voxels; L: left; R: right; x: Talairach coordinate x-axis; y: Talairach coordinate y-axis; z: Talairach coordinate z-axis.

**Table 7 brainsci-12-01714-t007:** Comparison of hemodynamic responses between the first and the last NF run (negative emotional-associated pictures minus neutral pictures; clusters of >30 voxels, q (FDR) < 0.001, T-score: 8 to −8).

MDD REAL								
			Centre of Gravity			Size	T-Score	
Brain Region	Side	BA	x	y	z		Ø	Max
(A) Last NF run vs. first NF run								
Medial Frontal Gyrus	L/R	10	0	53	3	541	4.95	7.20
Inferior Frontal Gyrus	L	47	−26	25	−4	446	5.35	7.41
Middle Temporal Gyrus	L	37	−45	−51	−6	471	5.05	6.13
Anterior Cingulate Gyrus	L	32/10	−19	44	8	610	4.85	6.15
(B) First NF run vs. last NF run								
Precentral Gyrus	R	6	30	−13	51	390	−4,70	−5.36
Middle Frontal Gyrus	R	6	35	14	47	610	−4.84	−6.05
	R	8	25	26	37	336	−4.29	−5.89
Medial Frontal Gyrus	R	9	18	38	31	462	−5.52	−8.04
	R	9	7	49	32	286	−5.31	−7.85
Inferior Frontal Gyrus	L	46	−44	40	7	471	−5.17	−6.65
Precuneus	R	31	15	−54	36	430	−4.66	−5.60
Parahippocampal Gyrus	R	19	38	−49	−2	309	−4.62	−5.28

Abbreviations: Brodmann area; side: hemisphere; x: Talairach coordinate x-axis; y: Talairach coordinate y-axis; z: Talairach coordinate z-axis; max: maximal T-score; Ø: average T-score; size: cluster size; voxels: number of activated voxels; L: left; R: right.

**Table 8 brainsci-12-01714-t008:** Comparison of hemodynamic responses between the first and the last NF run in the MDD REAL Responder group (negative emotion-associated pictures minus neutral pictures; clusters of >30 voxels, p (Bonf) < 0.05, T-score: 8 to −8).

MDD REAL Responder								
			Centre of Gravity			Size	T-Score	
Brain Region	Side	BA	x	y	z		Ø	Max
(A) Last NF run vs. first NF run								
Superior Frontal Gyrus	R	10	23.38	52.69	8.52	509	6.70	9.84
Superior/Medial Frontal Gyrus	L	6	−2.53	14.98	46.33	15899	6.58	15.38
Medial Frontal Gyrus	R	10	6.38	52.85	5.58	1368	7.71	15.35
	L	10	−3.87	53.7	5.03	863	13.33	7.59
	R	6	13.19	−17.48	57.34	313	5.65	6.52
Middle Frontal Gyrus	R	10	29.15	40.03	14.92	1127	6.38	9.81
Inferior Frontal Gyrus	R	47	49.78	26.55	−1.74	475	7.35	11.23
	L	47	−50.58	26.33	−0.41	248	6.53	9.11
Inferior Parietal Lobule/ Supramarginal Gyrus	R	40	54.52	−32.23	32.25	2184	6.77	12.60
	R	40	39.99	−43.94	33.17	439	5.65	6.73
	L	40	−55.31	−38.27	37.65	3607	6.86	13.51
	L	40	−52.59	−25.05	23.63	248	6.20	8.79
Anterior Cingulate	L	32	−14.42	37.26	8.65	3050	6.73	9.6
	R	24	6.51	35.18	10.95	2744	6.55	9.65
Cingulate Gyrus	L	24	−6.75	14.92	27.51	695	6.13	8.96
	R	24	6.97	11.52	28.45	931	6.19	8.86
Lingual Gyrus	R	19	31.46	−72.89	0.81	914	6.13	9.43
Inferior Occipital Gyrus	L	19	−34.66	−71.24	1.26	848	5.79	7.51
Cuneus	R	18	17.9	−84.39	17.06	399	5.89	8.46
Middle Temporal Gyrus	L	37	−46.06	−49.51	−4.56	1411	7.10	10.63
Insula	R	13	54.52	−32.23	32.25	2144	6.15	8.78
Lentiform Nucleus	R	-	16.32	−0.72	12.24	294	5.52	6.74
Caudate Head	L	-	−10.64	15.35	3.43	2538	7.18	13.05
Caudate Head	R	-	7.87	15.71	5.42	687	5.74	7.44
Putamen	L	-	−22.86	15.33	−2.11	1168	11.93	6.64
Medial Globus Pallidus	R	-	10.2	−6.59	−3.91	302	5.78	7.28
(B) First NF run vs. last NF run								
Middle Frontal Gyrus	R	8	33.05	18.09	48.66	317	−6.41	−9.96
	L	9	−44.32	21.07	31.84	6060	−7.37	−15.76
Superior Frontal Gyrus	R	9	8.33	55.01	23.95	279	−6.42	−9.96
	L	8	−0.87	38.72	45.9	1124	−8.76	−16.22
Lingual Gyrus	L	18	−7.32	−85.98	−0.82	765	−7.19	−12.11
Inferior Occipital Gyrus	L	19	−45.98	−75.25	−2.19	759	−7.27	−12.42
Middle Temporal Gyrus	L	21	−53.07	−17.9	−11.03	416	−5.91	−7.51
Parahippocampal Gyrus	R	19	37.44	−43.72	−0.4	836	−6.13	−8.24
Caudate Body	R	-	19.77	−18.58	28.87	1065	−6.00	−7.91

Abbreviations: BA: Brodmann area; side: hemisphere; x: Talairach coordinate x-axis; y: Talairach coordinate y-axis; z: Talairach coordinate z-axis; max: maximal t-score; Ø: average t-score; size: cluster size; voxels: number of activated voxels; L: left; R: right).

**Table 9 brainsci-12-01714-t009:** Comparison of hemodynamic responses between the first and the last NF run in the MDD REAL Non-Responder group (negative emotion-associated pictures minus neutral pictures; clusters of >30 voxels, q (FDR) < 0.001, T-score: 8 to −8).

MDD REAL Non-Responder								
			Centre of Gravity			Size	T-Score	
Brain Region	Side	BA	x	y	z		Ø	Max
(A) Last NF run vs. first NF run								
-	-	-	-	-	-	-	-	-
(B) First NF run vs. last NF run								
Middle Frontal Gyrus	R	6	36.84	10.36	42.84	363	−5.53	−6.18
Medial Frontal Gyrus	R	9	17.6	32.71	30.42	505	−5.51	−6.76
Inferior Parietal Lobule	L	40	34.11	−49.84	37.99	1523	−5.54	−6.90

Abbreviations: BA: Brodmann area; side: hemisphere; x: Talairach coordinate x-axis; y: Talairach coordinate y-axis; z: Talairach coordinate z-axis; max: maximal T-score; Ø: average T-score; size: cluster size; voxels: number of activated voxels; L: left; R: right.

**Table 10 brainsci-12-01714-t010:** Hemodynamic responses between the first and the last NF run (negative emotion-associated pictures minus neutral pictures; clusters of >30 voxels, q (FDR) < 0.001, T-score: 8 to −8).

			Centre of Gravity			Size	T-Score	
Brain Region	Side	BA	x	y	z		Ø	Max
HC REAL > MDD patients								
Middle Temporal Gyrus	R	39	42	−53	10	739	−4.94	6.14
MDD patients > HC REAL								
Inferior Parietal Lobule	R	40	58	−29	30	424	5.57	7.54
Supramarginal Gyrus	L	40	−57	−42	34	298	4.71	5.31
Parahippocampal Gyrus	L	19/36	−23	−45	−5	270	5.03	6.36
Lateral Globus Pallidus	L	-	−20	−4	−2	702	4.96	6.29

Abbreviations: BA: Brodmann area; side: hemisphere; x: Talairach coordinate x-axis; y: Talairach coordinate y-axis; z: Talairach coordinate z-axis; max: maximal T-score; Ø: average T-score; size: cluster size; voxels: number of activated voxels; L: left; R: right.

## Data Availability

The datasets generated for this study are available on request to the corresponding author.

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
