# Peer review of "Individual- and Connectivity-Based Real-Time fMRI Neurofeedback to Modulate Emotion-Related Brain Responses in Patients with Depression: A Pilot Study"

_brainsci, 2022, doi:10.3390/brainsci12121714_

Round 1

Reviewer 1 Report

The article “Individual and connectivity-based real-time fMRI neurofeedback to modulate emotion-related brain responses in patients with depression” explores an innovative approach of connectivity-based rtfMRI NF in patients with depression and shows that targeting connectivity between the dlPFC and the insular cortex is doable in a depression sample and healthy control sample.

My main comment would be that the interpretations of the results are very exhaustive and I think that the manuscript would gain clarity if the authors could prioritize more clearly between the different interpretations available according to their beliefs of what seems to be the most valuable option.

My minor comments are the following:

Because the abstract should be standalone information, I would have liked that the terms performer and non-performer group would be defined there to clear out all ambiguity.

In the introduction, I would also have appreciated a better definition of what the authors call “behavioral group therapeutic strategies”.

Since age and sex differences between samples are cited as limitations, I wonder if the authors could have integrated some work that investigated the impact of these variables or just state that this aspect of the literature is still uncharted and should be addressed in the future and emphasize the need of getting a group of matched healthy controls in the future.

The method section also left me curious about how connectivity was computed exactly (i.e., on how many TRs, using what type of correlation, and the rationale that led to those method decisions). I was also curious about how much real-time feedback and post-doc connectivity were similar, that would be a nice add-up to supplemental material if the authors have any.

I also wonder if the authors tried to differentiate between the performer and non-performer groups in the healthy control group and whether there were any differences between them. And in both groups, if the authors tried to investigate if some sociodemographic or clinical variable could drive the propensity of participants to effectively modulate the connectivity between both ROIs.

In the discussion, the sentence “In that context, an increase of neuronal activity in the mOFC could be a hint that NF is effective in depression patients” should be reworded in my opinion, given that with no impact of this intervention on BDI scores in the patient group, it is hard to talk about the effectiveness of this intervention per se, I think learning processes would be a better wording to mention this effect.

I also think the fact that this study is more of a pilot study or proof of concept should be mentioned in the title of the manuscript and the abstract.

Author Response

Comments and Suggestions for Authors
The article “Individual and connectivity-based real-time fMRI neurofeedback to modulate emotion-related brain responses in patients with depression” explores an innovative approach of connectivity-based rtfMRI NF in patients with depression and shows that targeting connectivity between the dlPFC and the insular cortex is doable in a depression sample and healthy control sample. My main comment would be that the interpretations of the results are very exhaustive and I think that the manuscript would gain clarity if the authors could prioritize more clearly between the different interpretations available according to their beliefs of what seems to be the most valuable option.
Answer: We thank the reviewer for this important note. As you wrote yourself, it is a pilot study and that’s why we wanted to keep the discussion more open for interpretations. However, following the suggestion, we have rewritten and shortened parts of the discussion. In this way, we concentrate more on the most important interpretations and hope to provide more clarity. We hope you will agree with it.
My minor comments are the following:
Because the abstract should be standalone information, I would have liked that the terms performer and non-performer group would be defined there to clear out all ambiguity.
Answer: We thank the reviewer for this valuable comment. The information about the performer/non-performer group was included in the abstract and we changed the group names to Responder and non-Responder group.
‘The MDD REAL group was divided into a Responder- and a non-Responder group. Patients with an increased connectivity during the second NF session or during both the first and the second NF session have been identified as “MDD REAL Responder” (N = 6). Patients that did not show any increase in connectivity and/or a decreased connectivity have been identified as “MDD REAL non-Responder” (N = 7).’
In the introduction, I would also have appreciated a better definition of what the authors call “behavioral group therapeutic strategies”.
Answer: We apologize for this unclear wording. The sentence has been rewritten:
‘To our knowledge, none of the previous studies in patients with major depression applied a connectivity-based NF training in the context of therapy within a psychosomatic day clinic.’
Since age and sex differences between samples are cited as limitations, I wonder if the authors could have integrated some work that investigated the impact of these variables or just state that this aspect of the literature is still uncharted and should be addressed in the future and emphasize the need of getting a group of matched healthy controls in the future.
Answer: Unfortunately we couldn’t find any specific study on fMRI neurofeedback and sex or age differences.
The method section also left me curious about how connectivity was computed exactly (i.e., on how many TRs, using what type of correlation, and the rationale that led to those method decisions). I was also curious about how much real-time feedback and post-hoc connectivity were similar, that would be a nice add-up to supplemental material if the authors have any.
Answer: We completely agree about the need to offer more information concerning the NF connectivity calculation and thank the reviewer for pointing this out. As the connectivity plugin was a completely new at the time of our study planning (beta-version of the Turbo-BrainVoyager developer) and experimental measurements prior the study using standard parameters showed feasible results, we used the given default settings recommended by the developer: online analysis
was based on Pearson’s correlation coefficient, and a sliding window of 20 TRs for the continuous calculation including baseline and NF blocks according to our design. We added this information in the respective ‘Materials and Methods’ sections (Page 8).
Unfortunally, we did not perform an analogous connectivity control run without NF after the NF runs that could be directly compared. We agree that this might have been very interesting. We will consider that in future studies.
I also wonder if the authors tried to differentiate between the performer and non-performer groups in the healthy control group and whether there were any differences between them.
Answer: Thank you very much for this suggestion: in the present study the main aim was to examine rtfMRI NF related changes in the emotion-related areas in patients with depression. In addition, we aimed at assessing the influence of rtfMRI NF on clinical data in patients with major depression. The differentiation of functional responses in Responders and non-Responders aimed at the examination if or to what extent functional responses can be addressed to the success during connectivity-modulation. However, we are totally aware that these results are preliminary given the small sample sizes of each subgroup and the missing of a clear hypothesis regarding differences between these subgroups (Responder/non-Responder).
Healthy subjects served as a control group. Given the fact that we did not present any hypothesis for functional variations within this group and the effect of Responders/non-Responders we decided to focus on the effects within the experimental group (patients with depression).
And in both groups, if the authors tried to investigate if some sociodemographic or clinical variable could drive the propensity of participants to effectively modulate the connectivity between both ROIs.
Answer: Thanks for that suggestion. We implemented a correlation between sex/age and BDI in the MDD Real-Perfomer group. There was no significant correlation.
In the discussion, the sentence “In that context, an increase of neuronal activity in the mOFC could be a hint that NF is effective in depression patients” should be reworded in my opinion, given that with no impact of this intervention on BDI scores in the patient group, it is hard to talk about the effectiveness of this intervention per se, I think learning processes would be a better wording to mention this effect.
Answer: We apologize for this unclear wording. The sentence has been rewritten:
‘In that context, an increase of neuronal activity in the mOFC could be a hint for NF-associated learning processes in depression patients.’
I also think the fact that this study is more of a pilot study or proof of concept should be mentioned in the title of the manuscript and the abstract.
Answer: We totally agree and adapted the manuscript (title, abstract, interpretation) accordingly.

Reviewer 2 Report

Overview and general recommendation:

This manuscript described a study to investigate whether the connectivity-based neurofeedback protocol was effective in patients with depression. In particular, what subjects needed to do was try to increase the functional connectivity between dlPFC and insula while watching some negative emotion related pictures. The real treatment was applied to both depression patients and healthy control subjects, but there was a lack of data from the sham condition. The study is novel in that the target of the neurofeedback is the functional connectivity between brain regions that have been found important in depression symptoms, instead of focusing on the brain activation levels on certain brain regions themselves. The experiment was thoughtfully designed, data analysis was carefully conducted, and results were properly reported.

Comments:

1.     Since the main result was only based on the real group, not including the sham group, I think it’s not necessary to describe the sham group at the top of page 5. It induced expectation that was not met when reading the paper. Accordingly, there is no need to report information about sham subjects in Table 1.

2.     It is less appropriate to address BOLD response as neuronal responses throughout the manuscript. BOLD responses reflect hemodynamic responses and they might be correlated with neuronal responses through some intermediate transformations. It would be misleading to call those measures as neuronal responses. On page 9, “neurobiological response” was used. I think using “brain response” in general or “hemodynamic response” would make more sense.

3.     It will be helpful to include a figure that shows the individual location of the dlpfc and insula ROIs across subjects on a normalized brain space. Related to this, why do both ROIs show up only on the left hemisphere? Does that correspond to any previous literature? Will this have any potential influence on the experimental effect? It would be better to include a short discussion paragraph on this.

4.     MDD real performer vs. MDD real non-performer. I failed to follow how the authors split the MDD real subjects into “performer” and “non-performer”. It was written on page 9 that “Patients with an increased connectivity during the second NF session or during both the first and the second NF session …” but it is unclear which connectivity was used here and what threshold was used to decide the cut-off.

5.     In section 3.2 Correlations between ROIs and questionnaires before NF. Other correlation results should also be reported, even if these are not significant. Specifically, the pairwise correlations between all questionnaire items and different brain regions should be included.

Author Response

Overview and general recommendation:
This manuscript described a study to investigate whether the connectivity-based neurofeedback protocol was effective in patients with depression. In particular, what subjects needed to do was try to increase the functional connectivity between dlPFC and insula while watching some negative emotion related pictures. The real treatment was applied to both depression patients and healthy control subjects, but there was a lack of data from the sham condition. The study is novel in that the target of the neurofeedback is the functional connectivity between brain regions that have been found important in depression symptoms, instead of focusing on the brain activation levels on certain brain regions themselves. The experiment was thoughtfully designed, data analysis was carefully conducted, and results were properly reported.
Comments:
1. Since the main result was only based on the real group, not including the sham group, I think it’s not necessary to describe the sham group at the top of page 5. It induced expectation that was not met when reading the paper. Accordingly, there is no need to report information about sham subjects in Table 1.
Answer: Thank you very much for this suggestion: we excluded the information about the sham group of table 1; in addition parts of the ‘subjects’ paragraph were rewritten and adapted to be more clear.
2. It is less appropriate to address BOLD response as neuronal responses throughout the manuscript. BOLD responses reflect hemodynamic responses and they might be correlated with neuronal responses through some intermediate transformations. It would be misleading to call those measures as neuronal responses. On page 9, “neurobiological response” was used. I think using “brain response” in general or “hemodynamic response” would make more sense.
Answer: We apologize for this unclear wording. The manuscript was adapted accordingly.
3. It will be helpful to include a figure that shows the individual location of the dlpfc and insula ROIs across subjects on a normalized brain space. Related to this, why do both ROIs show up only on the left hemisphere? Does that correspond to any previous literature? Will this have any potential influence on the experimental effect? It would be better to include a short discussion paragraph on this.
Answer: We really appreciate this comment as the reviewer focus on an important issue that we missed discussing and that shows a strength of our study design in our opinion. Therefore, we added the following paragraph in the discussion section (page 18) pointing out several aspects of this issue:
‘Moreover, a remarkable similarity across groups was revealed in the online-analyses of the emotion-associated task before the NF training (functional localizer). As stated in the ‘Materials and Methods’ section, almost all ROIs were located in the left hemisphere, i.e. in the left insula and the left dlPFC and not in the respective areas of the right hemisphere - determined using pictures with a negative emotional aspect. The issue of lateralization in the brain’s emotional processing including processing
of negative emotional visual stimuli has been conversely discussed in the past decades. Up to now, it is still unclear to which degree and at which level each hemisphere contributes to the processing of emotional information – however, activity in the left hemisphere is thought to be mainly involved in the regulation of negative emotions (90). Given the consistency of the left-accentuated BOLD response, our paradigm and setting seem to be suitable to reveal possible functional effects via similar functional pathways across subjects and to generate suitable ROIs for NF tasks. Moreover, the insula and the dlPFC of the left hemisphere are explicitly known to be affected in patients with MDD (91-92) and to be potential targets for rt-fMRI-NF (93).’.
4. MDD real performer vs. MDD real non-performer. I failed to follow how the authors split the MDD real subjects into “performer” and “non-performer”. It was written on page 9 that “Patients with an increased connectivity during the second NF session or during both the first and the second NF session …” but it is unclear which connectivity was used here and what threshold was used to decide the cut-off.
Answer: We apologize for this unclear wording. The sentence has been added:
‘A decision was not made according to a specific cut-off or average values, but according to a relative difference in connectivity between NF1 and NF3 on day 1 and day 2 respectively.’
5. In section 3.2 Correlations between ROIs and questionnaires before NF. Other correlation results should also be reported, even if these are not significant. Specifically, the pairwise correlations between all questionnaire items and different brain regions should be included.
Answer: Thanks for that suggestion. We implemented further correlation, see ‘Results section’, page 10.